# The Dose-Related Efficacy of Human Placenta-Derived Mesenchymal Stem Cell Transplantation on Antioxidant Effects in a Rat Model with Ovariectomy

**DOI:** 10.3390/antiox12081575

**Published:** 2023-08-07

**Authors:** Jin Seok, Hyeri Park, Dae-Hyun Lee, Jun Hyeong You, Gi Jin Kim

**Affiliations:** 1Department of Bioinspired Science, CHA University, Seongnam-si 13488, Republic of Korea; jjin8977@gmail.com (J.S.); plabrnd.park@gmail.com (H.P.); plabrnd.lee@gmail.com (D.-H.L.); yjh950210@gmail.com (J.H.Y.); 2PLABiologics, Co., Ltd., Seongnam-si 13522, Republic of Korea

**Keywords:** ovariectomized rat model, placenta-derived mesenchymal stem cells, ovary, folliculogenesis, antioxidants

## Abstract

Oxidative stress initiates various degenerative diseases, and it is caused by excessive reactive oxygen species (ROS) production. Oxidative stress is a key factor that causes infertility by inducing ovarian dysfunction, characterized by irregular hormone levels, lower quality of mature follicles, and loss of follicles. Hence, stem cell therapy has been actively studied as an approach to overcome the side effects of hormone replacement therapy (HRT) on ovarian dysfunction. However, there is a lack of evidence about the appropriate number of cells required for stem cell therapy. Therefore, based on the antioxidant effects investigated in this study, we focused on determining the appropriate dose of stem cells for transplantation in an animal model with ovarian dysfunction. One week after half-ovariectomy, placenta-derived mesenchymal stem cells (PD-MSCs, 1 × 10^5^ cells, 5 × 10^5^ cells, or 2.5 × 10^6^ cells) were injected intravenously into the Tx groups through the tail vein. As a result, the mRNA expression of hAlu gradually increased as the transplanted cell concentration increased. Compared with no transplantation (NTx), the transplantation of PD-MSCs improved folliculogenesis, including the levels of secreted hormones and numbers of follicles, by exerting antioxidant effects. Also, the levels of oxidized glutathione in the serum of animal models after transplantation were significantly increased (* *p* < 0.05). These results indicated that PD-MSC transplantation improved ovarian function in half-ovariectomized rats by exerting antioxidant effects. According to our data, increasing the number of transplanted cells did not proportionally increase the effectiveness of the treatment. We suggest that low-dose PD-MSC transplantation has the same therapeutic effect as described in previous studies. These findings provide new insights for further understanding reproductive systems and provide evidence for related clinical trials.

## 1. Introduction

Reactive oxygen species (ROS) are generated by various factors and act as signaling molecules that regulate physiological and biological processes. ROS are strongly associated with damage to lipids, proteins, and DNA, and they are the principal cause of oxidative stress; ROS include superoxide anions, hydrogen peroxide, and hydroxyl radicals. H_2_O_2_ is generated from superoxide, which is produced by mitochondria, endoplasmic reticulum (ER), cytosol, and NADPH oxidase. H_2_O_2_ is neutralized through the defense mechanisms of various antioxidant enzymes, which also include superoxide dismutase (SOD) and catalase rather than only glutathione [1]. In the ovary, ROS-induced oxidative stress mediates toxicant-induced destruction of ovarian follicles, which is currently studied using models established via exposure to various chemicals (e.g., cyclophosphamide, 4-vinyl cyclohexene epoxide, ionizing radiation, and H_2_O_2_). ROS are also produced by normal intracellular metabolic processes; however, if the generated ROS are not scavenged in time, their accumulation causes oxidative stress, which disrupts the homeostasis of oxidation and reduction processes [2]. Depending on the levels of these ROS, meiotic cell cycle progression and cell death affect can oocyte physiology [3,4]. Additionally, ROS levels are closely correlated with fertilization rates and reproductive outcomes. Based on previous reports, a moderate level of ROS stimulates the resumption of oocyte meiosis from diplotene as well as M-II arrest. However, a high level of ROS induces cell death (e.g., apoptosis and necrosis) and causes meiotic cell cycle arrest [5,6]. Granulosa cells are very sensitive to ROS, which can cause granulosa cell death. In follicle fluid, macrophages can generate high levels of ROS, which may trigger the cell death of oocytes as well as that of surrounding granulosa cells because ROS can cross cell membranes to affect surrounding cells. These findings suggest that the level of ROS and ROS regulation play important roles in ovarian function.

Mesenchymal stem cells (MSCs) have been highlighted in the fields of sepsis, degenerative disease, autoimmune disease, and transplant medicine because they have advantageous properties, including self-renewal, multilineage differentiation, immunomodulation, and cytokine and growth factor secretion. Moreover, the mode of action (MOA) of ROS has been demonstrated to exert proliferation, antiapoptosis, anti-inflammatory, and antioxidant effects [7]. Recently, among MSCs, human placenta-derived MSCs (PD-MSCs) have been reported to have the potential for the treatment of different degenerative diseases (e.g., cirrhosis of the liver, chemically and surgically induced ovarian dysfunction, and skin wounds) [8,9,10]. The reason for this potential is that these cells have beneficial therapeutic effects, such as no immune rejection and no ethical problems, compared with other types of MSCs. In particular, the antiapoptotic effects of MSCs are the mechanism underlying the recovery of organ function in various diseases. Many researchers have suggested that the paracrine effect of MSCs via the production of several cytokines is a powerful mechanism through which therapeutic effects can be achieved [11]. Previous reports have demonstrated that PD-MSCs can restore ovarian function, including reproductive hormone production and follicular development, by stimulating proliferation, vascular remodeling, and antioxidant effects [9,12,13].

To support the development of stem cell therapy based on these findings, more than 500 clinical trials are registered on ClinicalTrials (www.clinicaltrials.gov; accessed on 10 January 2023) that aim to explore the effect of stem cells in nearly every clinical application, including the treatment of neurodegenerative and cardiac disorders, graft-versus-host disease, perianal fistulas, etc. In addition, several preclinical studies have attempted to improve ovarian dysfunction, and clinical trials that aim to develop treatments have also been reported on this website (www.clinicaltrials.gov). Although many clinical trials on the use of stem cell therapy to treat premature ovarian failure (POF) have been registered, these studies have many limitations due to a lack of clear preclinical results and clinical trial progress [14]. The reason for the poor development of treatments is the absence of clear suggestions for treatment development, the lack of specific standardization based on effective MOAs as well as the number of cells needed, and the optimal injection route. In one of the reported preclinical trials, Wang and colleagues investigated the effect of UC-MSC transplantation with cell concentrations of 0.25 × 10^6^ cells/mL, 1.00 × 10^6^ cells/mL, and 4.00 × 10^6^ cells/mL via the tail vein in autoimmune-induced POF rats. These authors reported that the effect of umbilical cord mesenchymal stem cell (UC-MSC) transplantation increased with an increase in cell dose [15]. However, research on the relationship between cell dose and therapeutic effects is rare, making the results uncertain; thus, studies on the dose and type of stem cells in various ovarian dysfunction models are needed. On this basis, although there are few studies about the efficacy of different stem cell concentrations in models of ovarian dysfunction, including degenerative diseases, the determination of a proper stem cell concentration to achieve therapeutic efficacy is important and necessary for the development of stem cell treatments [16,17].

Although we reported the therapeutic effects of PD-MSCs in rats with ovarian dysfunction, there is still a lack of evidence regarding the optimal cell dose for the development of stem cell therapy for ovarian failure disease. Hence, our aims in this study were to compare the efficacy of the transplantation of three different cell concentrations and to determine whether ovarian function improves through key MOAs, such as antioxidant effects, in rats with ovarian dysfunction induced by performing surgical procedures other than ovariectomy.

## 2. Methods

### 2.1. Cell Culture of PD-MSCs

With the approval of the Institutional Review Board (IRB) of CHA General Hospital (IRB07-18; Seoul, Korea), human PD-MSCs were isolated from the chorionic plate membrane of placental tissues, which is a fetal side of the placenta, that were collected at term (38 ± 2 gestational weeks) after obtaining patient consent for the use of their stem cells for research, as previously described by Lee et al. Briefly, PD-MSCs were cultured in alpha-minimum essential medium (α-MEM; HyClone Laboratories Inc., South Logan, UT, USA) supplemented with 10% fetal bovine serum (FBS; GIBCO-BRL, Waltham, MA, USA), 1% penicillin–streptomycin (GIBCO-BRL), 25 ng/mL fibroblast growth factor 4 (FGF4; Peprotech, Rocky Hill, NJ, USA), and 1 μg/mL heparin (Sigma-Aldrich, Munich, Germany) at 37 °C in a humidified atmosphere containing 5% CO_2_ and 95% air. Before transplantation, PD-MSCs were stained with PKH67 dye (PKH fluorescent cell linker kit; Sigma-Aldrich) to monitor engraftment.

### 2.2. Establishment of a 1/2 OVX Rat Model

All the animal experiments were approved by the Institutional Animal Care and Use Committee (IACUC-190083) of the CHA Laboratory Animal Research Center (Gyeonggi-do, Korea). In this study, 7-week-old female Sprague Dawley rats (Orient-Bio Inc., Seongnam-si, Korea) were used to establish models of ovarian failure. During ovariectomy, the rats were fully anesthetized using avertin (2,2,2-tribromoethanol, Sigma-Aldrich). Half of each ovary was excised through skin and muscle incisions. After the excision of half of each ovary, Vetbond^TM^ tissue adhesive (3M, Paul, MN, USA) was added to prevent bleeding and dehydration. Then, the surgical site was sutured and disinfected with povidone-iodine (Sigma-Aldrich). One week after half-ovariectomy, PD-MSCs (1 × 10^5^ cells, 5 × 10^5^ cells, or 2.5 × 10^6^ cells; passages 11–13) were injected intravenously into the Tx groups through the tail vein. All the rats were sacrificed after 1, 3, and 5 weeks, and ovarian tissues and blood samples were collected.

### 2.3. Ovarian Explant Culture (Ex Vivo)

To compare PD-MSC efficacy with in vivo experiments, we studied ovarian explant culture with PD-MSC cocultivation. After the dissected ovarian tissues were embedded in Matrigel-coated (BD science) wells, PD-MSCs were added at densities that were proportional to the numbers administered in vivo. In brief, 2 × 10^3^ PD-MSCs were added to the coculture system to model the 1 × 10^5^ Tx group, 1 × 10^4^ PD-MSCs were added to model the 5 × 10^5^ Tx group, and 5 × 10^4^ PD-MSCs were added to model the 2.5 × 10^6^ Tx group; these coculture systems were established using a 24-well insert system (BD Science) and incubated for 24 and 48 h. Next, the mRNA expression patterns of the harvested ovarian tissues were analyzed to assess follicular development and antioxidant activity.

### 2.4. RNA Isolation and Quantitative Polymerase Reaction

To analyze mRNA expression, the total RNA was extracted from homogenized ovarian tissues using TRIzol reagent (Ambion, Boston, MA, USA) according to the manufacturer’s protocol. Then, cDNA was synthesized using Superscript III reverse transcriptase (Invitrogen Corporation, Waltham, MA, USA). The conditions of cDNA synthesis were as follows: 5 min at 65 °C, 1 min at 4 °C, 60 min at 50 °C, and 15 min at 72 °C. The cDNA was used for qRT–PCR analysis, which was performed with SYBR Green Master Mix (Roche Holding AG, Basel, Switzerland). The primer sequences that were used are listed in Appendix A. Rat GAPDH was used as an internal control for normalization, and each sample was analyzed in triplicate.

### 2.5. Protein Isolation and Western Blotting Analysis

To analyze protein expression, the homogenized ovarian tissues were lysed with radioimmunoprecipitation assay buffer (RIPA; Sigma-Aldrich) supplemented with a phosphatase inhibitor cocktail (genDEPOT, Katy, TX, USA) and protease inhibitor (Roche Holding AG). The concentrations of the lysates were quantified by using a bicinchoninic assay kit (BCA; Thermo Fisher, Waltham, MA, USA), and the concentrations were normalized. Equal amounts of lysates of each group were separated using sodium dodecyl sulfate–polyacrylamide gel electrophoresis (SDS–PAGE) according to protein molecular size. The separated proteins were transferred to polyvinylidene difluoride membranes (PVDF; Bio-Rad Laboratories, Hercules, CA, USA) using a Turbo system (Bio-Rad Laboratories). Then, the membranes were blocked with blocking buffer supplemented with 5% bovine serum albumin (BSA; Morebio, Gyeonggi-do, Korea) at room temperature for 1 h and incubated with the primary antibody in 2% BSA at 4 °C overnight. The primary antibody that was used included an anti-catalase antibody (ab52477; Abcam, Cambridge, UK) diluted at 1:500 and an anti-GAPDH antibody (LF-PA0018; ABfrontier, Seoul, Korea) diluted at 1:5000. After incubation, the membranes were washed with 1× Tris-buffered saline (TBS; eLbio, Gyeonggi-do, Korea) supplemented with Tween-20 (Biosolution, Pusan, Korea) and incubated with secondary antibody in 2% BSA at room temperature for 1 h. Protein expression was visualized using a Chemi Dox XRS+ imaging system (Bio-Rad Laboratories). The fold change in band intensity was used to compare target gene expression in the ovary.

### 2.6. Enzyme-Linked Immunosorbent Assay

All the rat serum samples were extracted from rat whole blood using a Vacutainer SST II plus plastic serum tube (BD Vacutainer; BD Sciences, San Jose, CA, USA) at 1300 RCF for 15 min. All the individual samples were analyzed to measure the levels of E2 (Biovision, Milpitas, CA, USA), AMH (Elabioscience Biotechnology, Houston, TX, USA), FSH (Abnova, Taipei, Taiwan), oxidized glutathione (GSSG; Arbor assay, Ann Arbor, MI, USA), and LDH (Cusabio Technology, Houston, TX, USA) with kits andfollowing manufacturers’ instructions. All the samples were analyzed three times, and the results are presented as the relative concentration in serum.

### 2.7. Immunofluorescence Staining for the Accumulation of Superoxide in Mitochondria

After the ovarian tissues were sectioned from frozen blocks, they were washed in 1 × phosphate-buffered saline (PBS; eLbio). Then, the sectioned tissues were incubated with 1.5 μm Mito SOX (Superoxide staining, red signals; Invitrogen Corporation) and 50 nM Mito Tracker (Mitochondria staining, green signals; Invitrogen Corporation) for 40 min at 37 °C. After staining, the sectioned tissues were washed and mounted by using Vectashield antifade mounting medium with DAPI (Vector Labs, Burlingame, CA, USA). The slides were observed via confocal microscopy (Zeiss 780; Zeiss, Oberkochen, Germany) at 200× magnification, and images of randomized areas of all the slides were captured.

### 2.8. Histological Analysis of the Number of Follicles

To analyze the number of follicles according to the follicle maturation stage, ovarian tissues were fixed with 10% neutral buffered formalin (NBF; BBC Biochemical Corporation, McKinney, TX, USA) and embedded in paraffin. For histological analysis, the tissues were sectioned, with four tissues per slide, at a thickness of 4 μm. Then, the slideswere selected at 100 μm thickness intervals and stained using hematoxylin (Dako, Glostrup, Denmark) and eosin Y (BioGnost, Zagreb, Croatia). All the stained slides were scanned and analyzed using a Panoramic scanner (3D HISTECH Ltd., Budapest, Hungary).

### 2.9. Histological Analysis for Immunohistochemistry and TUNEL Assay

To analyze proliferation using PCNA expression, ovarian tissue sections that were cut from the paraffin blocks were deparaffinized. After hydrogen peroxide and antigen retrieval, the ovarian tissues were incubated with a blocking solution (Dako) for 1 h. Then, the ovarian tissues were incubated in a diluent buffer containing primary antibody (PCNA; Santa-Cruz, Dallas, TX, USA); 1:2000) at 4 °C overnight. After the ovarian tissues were washed, they were incubated with 1 h secondary antibody (Envision) at room temperature for 1 h. After washing, we used the DAB developing kit (Envision). Finally, all the ovarian tissues were counterstained with hematoxylin for 1 min and dehydrated. All the stained slides were scanned and analyzed using a Panoramic scanner (3D HISTECH Ltd.). To quantify the intensity of follicle staining, we measured the intensity of follicle staining according to development using the Dongle software program (3D HISTECH Ltd.). Three animals from each group were analyzed.

To analyze apoptosis using DNA fragmentation, all ovarian tissues were stained with a TUNEL assay kit (Abcam) according to the manufacturer’s protocol. Briefly, all the ovarian tissues were rehydrated and permeabilized with proteinase K. After washing the slides with 1 × TBS for 5 min, the ovarian tissues were exposed to 3% H_2_O_2_ to inactivate endogenous peroxidases. Then, all the ovarian tissues were stained with DAB solution after labeling using TdT equilibration buffer and TdT enzyme. To quantify the positive signals in the ovarian follicles, all the stained slides were scanned and analyzed using a Panoramic scanner (3D HISTECH Ltd.), and the staining intensity was measured using the Dongle software program (3D HISTECH Ltd.). Data refer to the staining intensity of the antral follicles. Three animals from each group were analyzed.

### 2.10. Statistical Analysis

All the experiments were analyzed in triplicate. All the results were analyzed by performing a one-way ANOVA using GraphPad Prism and are presented as the mean ± standard error (SE). Additionally, GraphPad Prism was used to analyze groupwise comparisons, and *p* values less than 0.05 indicated statistical significance.

## 3. Results

### 3.1. Transplanted PD-MSCs Can Engraft into Injured Ovarian Tissues in a Half-Ovariectomized Model

Before starting this study, to establish an animal model, we conducted a comparative analysis of the functional part of the hemi-OVX and 1/2 OVX models. Briefly, hemi-OVX, which was used in this study, is a model in which only one ovary is removed, and the other ovary is maintained, whereas 1/2 OVX is a model in which only half of both ovaries are removed. The reason for using this model is that if both ovaries are removed like the existing OVX model, it is possible to analyze the ovarian dysfunction model, but the actual ovarian function cannot be analyzed.

To confirm the establishment of an ovarian failure rat model using a method other than ovariectomy, we analyzed the mRNA expression of the genes related to folliculogenesis in the ovarian tissues of 1/2 OVX rats. The mRNA expression levels of Nanos C2HC-type zinc finger 3 (Nanos3), newborn ovary homeobox gene (Nobox), Lim homeobox 8 (Lhx8), and Lin28 homology A (Lin28a) were significantly decreased in the ovarian tissues of both the hemi-OVX and 1/2 OVX rats 5 weeks after surgery, compared with the normal rats. Additionally, these expression levels were significantly decreased in the ovarian tissues of the 1/2 OVX rats, compared with those of the hemi-OVX rats (Appendix AA–D; *p* < 0.05). These data indicate that 1/2 OVX rats experience more severe ovarian failure than hemi-OVX rats.

Generally, the ratio of ovary weight to body weight indicates ovarian reserveand allows forequal incisions in the ovarian tissues. In our data, the ovary weight in the nontransplantation (NTx) and transplantation (Tx) groups of the half-ovariectomized (1/2 OVX) group was significantly decreased compared with that in the normal group. However, the ratio was not different between the NTx and Tx groups (Figure 1A; *p* < 0.05). To confirm engraftment, we analyzed the mRNA expression of the human-specific Alu gene and the numbers of PKH67-labeled PD-MSCs in the ovarian tissues of 1/2 OVX rats 1 week after surgery. The mRNA expression of the hAlu gene was significantly increased in the 1 × 10^5^, 5 × 10^5^, and 2.5 × 10^6^ Tx groups, compared with the normal group, and in the 5 × 10^5^ and 2.5 × 10^6^ Tx groups, compared with the 1 × 10^5^ Tx group. However, there was no significant difference between the 5 × 10^5^ and 2.5 × 10^6^ Tx groups (Figure 1B; *p* < 0.05). Positive signals of engraftment were observed in the theca layer of follicles in the 1 × 10^5^, 5 × 10^5^, and 2.5 × 10^6^ Tx groups compared with the NTx group. Additionally, these signals were markedly increased in the 1 × 10^5^, 5 × 10^5^, and 2.5 × 10^6^ Tx groups in a cell-concentration-dependent manner (Figure 1C; *p* < 0.05). This finding suggests that the transplanted PD-MSCs engrafted into the injured ovarian tissues of 1/2 OVX rats.

### 3.2. Transplanted PD-MSCs Improve Ovarian Function with Follicular Development in 1/2 OVX Model

During follicular development, various genes are involved in the process of improving ovarian function. In the early stage, various genes, including Nanos C2HC-Type Zinc Finger 3 (Nanos3), lim-homeobox (Lhx8), and lin-29 homolog A (Lin28a), inducethe differentiation of primordial germ cells into primordial follicles in the ovary. The NOBOX oogenesis homeobox (Nobox) gene induces the differentiation of primordial follicles into primary follicles, the bone morphogenesis 15 (BMP15) gene is expressed during the process of primary follicle maturation for ovulation, and the epithelial growth factor receptor (EGFR) gene is related to antral follicle maturation for ovulation (Figure 2A). Hence, we analyzed the effect of transplanted PD-MSCs on follicular development using qRT–PCR. The mRNA expression of nanos3 in the ovarian tissues of 1/2 OVX rats was significantly decreased in the NTx group, compared with the normal group, and was significantly increased in the 1 × 10^5^, 5 × 10^5^, and 2.5 × 10^6^ Tx groups, compared with the NTx group. Additionally, this expression did not differ with the increase in transplanted cell concentrations (Figure 2B; *p* < 0.05). The mRNA expression of Lhx8, Lin28a, Nobox, BMP15, and EGFR in the ovarian tissues of 1/2 OVX rats was significantly decreased in the NTx group, compared with the normal group, and significantly increased in the 1 × 10^5^, 5 × 10^5^, and 2.5 × 10^6^ Tx groups, compared with the NTx group. However, there was no effect of transplanted PD-MSC concentration on follicular development (Figure 2B–G; *p* < 0.05). These data indicate that the transplanted PD-MSCs stimulated follicular development through folliculogenesis. Additionally, the folliculogenesis activity was the highest in the group transplanted with the lowest PD-MSC concentration compared with the other groups transplanted with other PD-MSC concentrations.

### 3.3. Transplanted PD-MSCs Regulate Ovarian Function, Increasing the Reproductive Hormone and Numbers of Follicles in the 1/2 OVX Model

To confirm ovarian function, we analyzed the hormone levels and follicle numbers in 1/2 OVX rats. The level of estrogen (E2) was increased in the NTx and 2.5 × 10^6^ groups, compared with the normal group, and was decreased in the 1 × 10^5^ and 5 × 10^5^ Tx groups, compared with the NTx group (Figure 3A). Additionally, the level of follicle-stimulating hormone (FSH) was increased in the NTx and 2.5 × 10^6^ Tx groups, compared with the normal group, and was decreased in the 1 × 10^5^ and 5 × 10^5^ Tx groups, compared with the NTx group (Figure 3B; *p* < 0.05). The level of anti-Mullerian hormone (AMH) was decreased in the NTx group, compared with the normal group, and was significantly increased in the 1 × 10^5^, 5 × 10^5^, and 2.5 × 10^6^ Tx groups, compared with the NTx group (Figure 3C; *p* < 0.05). These data indicate that transplanted PD-MSCs regulate the levels of reproductive hormones that are important for ovarian function.

To confirm that ovarian function was practically restored via PD-MSC transplantation, we counted the number of follicles in the ovarian tissues of 1/2 OVX rats. As shown in Figure 3D, the number of follicles, including primordial follicles, decreased in the NTx group compared with the normal group and increased in the 1 × 10^5^, 5 × 10^5^, and 2.5 × 10^6^ Tx groups compared with the NTx group (Figure 3D). Then, the numbers of follicles in each stage of follicular maturation were counted, and the results showed that the total follicles in ovarian tissues were decreased in the NTx group compared with the normal group and increased in the 1 × 10^5^, 5 × 10^5^, and 2.5 × 10^6^ Tx groups compared with the NTx group (Figure 3E, Table 1; *p* < 0.05). These data indicate that transplanted PD-MSCs improve the levels of reproductive hormones and the number of follicles during follicular development.

### 3.4. Transplanted PD-MSCs Stimulate Proliferation by Inhibiting Cell Death in a 1/2 OVX Rat Model

To analyze proliferation and cell death after transplantation, we performed immunohistochemical staining. As shown in Figure 4A, the area that was positive for proliferating cell nuclear antigen (PCNA) staining in the ovarian tissues was smaller in the NTx group than in the normal group. This area was larger in the 1 × 10^5^, 5 × 10^5^, and 2.5 × 10^6^ Tx groups than in the NTx group after PD-MSC transplantation (Figure 4A). Based on this result, we quantified the intensities of the PCNA signals in the antral follicles of ovarian tissues. The intensities of the positive signals were decreased in the NTx group, compared with the normal group, but after PD-MSC transplantation, these signals were increased in the 1 × 10^5^, 5 × 10^5^, and 2.5 × 10^6^ Tx groups compared with the NTx group (Figure 4B; *p* < 0.05). Hence, we confirmed cell death in ovarian tissues using TUNEL staining. The results showed that the positive signals indicating DNA damage were higher in the NTx group than in the normal group. After transplantation, these positive signals were decreased compared with that in the NTx group (Figure 4C). When these signals were quantified, the positive signals in the antral follicles were increased in the NTx group compared with the normal group and decreased in the 1 × 10^5^, 5 × 10^5^, and 2.5 × 10^6^ Tx groups, compared with the NTx group (Figure 4D; *p* < 0.05). Based on these results, we analyzed the level of lactate dehydrogenase (LDH), which is an indicator of cell death by necrosis in response to various inflammatory stimuli, in individual rat serum. The levels of LDH were significantly increased in the NTx group compared with the normal group and decreased in the Tx groups. Interestingly, the levels of LDH were markedly decreased in the 5 × 10^5^ and 2.5 × 10^6^ Tx groups compared with the 1 × 10^5^ Tx group (Figure 4E; *p* < 0.05). The gene expression of caspase 3 was much higher in the NTx group than in the normal group 1 week after surgery, and this gene expression was lower after PD-MSC transplantation (Appendix A). This finding suggests that transplanted PD-MSCs induce proliferation and inhibit cell death in antral follicles.

### 3.5. Transplanted PD-MSCs Enhance Antioxidant Activity to Reduce Oxidative Stress in the 1/2 OVX Model

ROS production and antioxidant enzyme activation occur in various organelles in the cell, such as the cellular membrane, ER, and mitochondria, via dynamic cellular metabolism. As shown in Figure 5A, NADPH oxidase 4 (NOX4) produces H_2_O_2_ through NAPDH oxidase, and this process also occurs in the ER and cellular membrane. These free radicals are scavenged by antioxidants, such as protein disulfideisomerase (P4hb), SOD-1/-2, and heme oxygenase-1/-2 (HO-1/-2). Additionally, this process is regulated by factors related to mitochondrial biogenesis, such as peroxisome proliferator-activated receptor gamma coactivator 1-alpha (PGC1a) and Sirtuin 7 (SIRT7) [18].

On this basis, we analyzed the mRNA expression of factors related to both oxidative stress and antioxidant responses in ovarian tissues. First, the mRNA expression of NOX4, which produces ROS such as superoxide by transferring electrons from NADPH to molecular oxygen, was significantly increased in the NTx and Tx groups, compared with the normal group. Among the Tx groups, the 1 × 10^5^ Tx group exhibited higher NOX expression than the other groups (Figure 5B; *p* < 0.05). The mRNA expression of Ph4b, which is in the ER and reduces mtROS, showed a similar trend as NOX4 mRNA expression. Ph4b mRNA expression was significantly increased in the NTx and Tx groups compared with the normal group. Among the Tx groups, the 1 × 10^5^ Tx group exhibited higher expression than the other groups (Figure 5C; *p* < 0.05). HO-1 and HO-2 respond to ROS and intermediate products via the antioxidant signaling pathway. In the NTx group, the mRNA expression of HO-1 was slightly decreased, and the mRNA expression of HO-2 was significantly decreased, compared with the normal group. However, the mRNA expression of HO-1 and HO-2 was higher in the 1 × 10^5^, 5 × 10^5^, and 2.5 × 10^6^ Tx groups than in the NTx group after PD-MSC transplantation (Figure 5D,E; *p* < 0.05). The mRNA expression of SOD1, which is an antioxidant defense molecule that is expressed in nearly all living cells exposed to oxygen, was significantly decreased in the NTx group compared with the normal group. Interestingly, the mRNA expression of SOD1 gradually increased with the increase in transplanted cell concentration in the Tx groups (Figure 5F; *p* < 0.05). However, the mRNA expression of SOD2 showed a trend that differed from the mRNA expression of SOD1. It was increased in the NTx and Tx groups compared with the normal group. Among the Tx groups, it was increased in the 1 × 10^5^ Tx group compared with the other groups (Figure 5G; *p* < 0.05). Then, we analyzed mitochondrial function to assess the changes in mitochondrial biogenesis that occur in response to antioxidant effects. As a result, the mRNA expression of SIRT7 and PGC1a was significantly decreased in the NTx group compared with the normal group. In contrast, the expression of these molecules in the 1 × 10^5^ Tx groups was significantly higher than that in the other groups (Figure 5H,I; *p* < 0.05). This finding suggests that the transplanted PD-MSCs reduced ROS levels and oxidative stress through the activation of antioxidant enzymes. Additionally, antioxidant activity was highest in the group that underwent transplantation with the lowest concentration of PD-MSCs compared with the groups that underwent transplantation with other concentrations of PD-MSCs.

### 3.6. Transplanted PD-MSCs Enhance Antioxidant Activity and Reduce Cell Death in a 1/2 OVX Model

To analyze the antioxidant effect of the transplanted PD-MSCs, we confirmed the levels of superoxide in the mitochondria. As shown in Figure 6A, the accumulated levels of superoxide (i.e., MitoSOX staining; red positive signal) in the mitochondria (i.e., MitoTracker staining; green positive signal) was dramatically higher in the NTx group than in the normal group. However, there were dose-dependent decreases in the 1 × 10^5^, 5 × 10^5^, and 2.5 × 10^6^ Tx groups compared with the NTx groupafter PD-MSC transplantation (Figure 6A). Based on this result, we quantified the superoxide ratio in the mitochondria. The accumulated superoxide ratio was significantly increased in the NTx group compared with the normal group and decreased in the 1 × 10^5^, 5 × 10^5^, and 2.5 × 10^6^ Tx groups compared with the NTx group after PD-MSC transplantation (Figure 6B; *p* < 0.05). These data indicated that the transplanted PD-MSCs attenuated the accumulation of superoxide in the mitochondria. Additionally, we analyzed the expression of genes related to oxidative stress and genes that encode antioxidant enzymes in ovarian tissues. The gene expression of HO-1 was not significantly different among the groups, but the gene expression of HO-2 was decreased in the NTx and Tx groups compared with the normal group. Among these groups, HO-2 gene expression in the 2.5 × 10^6^ Tx group was significantly decreased compared with that in the other groups (Appendix A; *p* < 0.05). Interestingly, the gene expression of SOD1 and catalase, which are final products of the antioxidant response, was significantly decreased in the NTx group compared with the normal group and slightly increased in the 1 × 10^5^, 5 × 10,^5^ and 2.5 × 10^6^ Tx groups compared with the NTx group after PD-MSC transplantation (Figure 6C,D; *p* < 0.05). Hence, we measured the levels of oxidized glutathione in individual rat serum samples using ELISA. Oxidized glutathione is converted from glutathione (GSH) to glutathione disulfide (GSSG) and plays important roles in numerous redox reactions, such as those involved in the detoxification of harmful substances and reactions that prevent oxidative damage [19]. The results showed that the levels of oxidized glutathione in individual rat serum samples were dramatically decreased in the NTx group compared with the normal group. Additionally, the levels of oxidized glutathione were significantly increased in the 1 × 10^5^, 5 × 10^5^, and 2.5 × 10^6^ Tx groups compared with the NTx group after PD-MSC transplantation (Figure 6E; *p* < 0.05). These data indicated that transplanted PD-MSCs inhibited the cell death signaling pathway by increasing the antioxidant potential.

### 3.7. PD-MSC Cocultivation Improves Antioxidant Activity and Reduces Cell Death in a 1/2 OVX Model

To confirm the effects of PD-MSCs on follicular development and antioxidant activity, we cocultured ovarian explants with PD-MSCs for 24 and 48 h. After PD-MSC cocultivation for 24 h, the effect on follicular development and antioxidant activity was weak. Interestingly, the mRNA expression of genes related to follicular development (e.g., Nanos3, Nobox, Lhx8, Lin28a, EGFR, and BMP15) was significantly increased in the PD-MSC cocultivation group compared with the non-cocultivation groups in a cell-dose-dependent manner at 48 h (Figure 7A–F; *p* < 0.05). Additionally, the mRNA expression of genes related to oxidative stress (e.g., NOX4 and HO1) was significantly decreased, and the mRNA expression of SOD1, which is an antioxidant enzyme, was significantly increased in the PD-MSC cocultivation groups compared with the non-cocultivation groups in a cell-dose-dependent manner at 48 h (Figure 7G–I; *p* < 0.05). The accumulated levels of superoxide in the mitochondria were higher in the 1 × PD-MSCs group than in the sham group at 48 h. However, there was dramatically lower in the 5 × and 25 × PD-MSCs cocultivation groups than the sham group (Figure 7A,B) at 48 h. This finding suggests that the coculture of PD-MSCs restores ovarian function by increasing antioxidant effects ex vivo.

## 4. Discussion

In the human body, moderate levels of ROS play indispensable roles in cellular metabolism, intracellular signaling, and defense against pathogens, whereas excessive levels of ROS accelerate the onset of diverse human diseases by disrupting defense systems [20]. Recently, antioxidants have been highlighted as medicines due to their ability to protect against oxidative cell damage, which is linked to chronic inflammatory diseases, such as Alzheimer’s disease, cardiac disease, and heart disease [21]. In regenerative medicine, MSCs modulate the immune response via the production of anti-inflammatory cytokines and many bioactive molecules. Moreover, it has been reported that MSCs can regulate oxidative-stress-induced inflammatory responses through their antioxidant capacity [22]. According to recent reports, MSCs exert antioxidant effects by suppressing inflammation, directly scavenging ROS, and inducing endogenous antioxidant defenses in various contexts (e.g., aging, reproduction and liver, vascular, and renal studies). Stavely and Nurgali demonstrated that MSC therapy is useful in regenerative medicine due to its antioxidant properties, considering that ROS, whose production is induced by oxidative stress, is implicated in almost all diseases [23]. Specifically, Alizadeh and colleagues suggested that cytokines such as hepatocyte growth factor (HGF), insulin-like growth factor (IGF), and pigment epithelium-derived factor (PEDF) secreted by MSCs show antioxidant effects under oxidative conditions [24]. Based on previous reports indicating that antioxidants play an important role in improving follicular development in rats with ovarian dysfunction, our data demonstrated that transplanted PD-MSCs attenuated oxidative stress by exerting antioxidant effects in a 1/2 OVX rat model as well as in OVX rats.

Follicle development is a dynamic process that precedes ovulation and occurs via the primordial, primary, secondary, and antral stages, resulting in the coordination of interactions between granulosa and theca cells through various growth factors. At the primordial follicle stage, oocytes are arrested under dictyotene conditions, and flattened ovarian follicular epithelial cells are formed in the layer around the oocyte. Currently, granulosa cells proliferate to prepare oocytes for maturation. After the early stage of follicular development, theca cells that are recruited by granulosa cells regulate the proliferation and apoptosis of granulosa cells in the antral follicle stage via the production of thecal factors [25]. The apoptosis of follicles, including atresia or atretic follicles, occurs during follicular development due to inadequate growth support and several external factors. Finally, ovulated eggs are the product of the appropriate interactions of granulosa and theca cells, including via the sufficient production of growth factors [26]. As such, the proliferation and apoptosis of follicular cells in the ovary are important parts of development. Our immunohistochemistry staining data showed that transplanted PD-MSCs induced proliferation and attenuated cell death in antral follicles. These findings indicate that PD-MSCs can promote follicular development by secreting growth factors.

As the number of research papers in which the therapeutic efficacy of stem cells has been reported in various diseases increases, several clinical trials are also being conducted. According to Fooladi’s reports, stem cells are currently being tested in diseases such as liver, cardiovascular, graft versus host, Crohn’s, multiple sclerosis, systemic lupus erythematosus, rheumatoid arthritis, type 1 diabetes, and cartilage diseases [14]. However, preclinical research and clinical trials that have been reported so far only show that stem cells have a therapeutic effect but do not provide clear guidelines for optimal cell capacity according to diseases and transplant cell types. Hence, to develop stem cell therapy, there is an urgent need for research on the number of cells needed for transplantation, the optimal route of administration, the type of stem cells that should be used, and the therapeutic effects of this approach. Also, the use of high doses of stem cells has raised many concerns regarding safety and side effects. The causes of this concern are the possibility of immune rejection, tumor formation due to the higher proliferative capacity of high doses of stem cells, and the development of idiopathic pulmonary fibrosis due to stem cell invasion after IV injection. Therefore, many clinical researchers have agreed that consistent results are needed to determine the optimal cell dose, including data about the safety and consistent therapeutic efficacy resulting from the administration of an appropriate cell dose. For this reason, our study was conducted to help determine the optimal cell dose.

For several diseases, in the case of osteoarthritis treated with AD-MSC in clinical trials, the high dose (5 × 10^7^) of stem cells resulted in greater improvement in knee function than the low (1 × 10^7^) and middle (2 × 10^7^) doses of stem cells [27]. Inthe reported studies that compared the effects of stem cells as cell dose increased in the treatment of cardiomyopathy, the findings were paradoxical, although the dose of transplanted stem cells differed according to the route of administration [28]. However, some studies have shown a direct relationship between cell dose and efficacy: The cell dose range used for direct epicardial administration is 20 million to 450 million, but there were differences in the cell dose at which efficacy was observed [29,30]. In the case of dermal regeneration with UC-MSC treatment, a low dose of stem cells (2–4 × 10^4^ cells/cm^2^) regenerated the full-thickness excised burn wounds and reduced macrophage numbers; these results were better than those achieved with the middle (2–4 × 10^5^ cells/cm^2^) and high (2 × 10^6^ cells/cm^2^) doses of stem cells [31].

For ovarian disease, we have summarized studies that have until recently demonstrated MSCs’ therapeutic effects in ovarian dysfunction animal models (e.g., mouse and rat). According to the summarized data inAppendix A, most studies have only dealt with therapeutic effects, and only a few studies have dealt with therapeutic effects according to the dose of transplanted MSCs. In Qi’s study, the improvement in ovarian function and therapeutic effects of stem cells in the ovarian dysfunction model was investigated according to stem cell capacity from low to high doses (2.5 × 10^5^, 1 × 10^6^, 4 × 10^6^). They confirmed that high doses of stem cells are more effective in improving ovarian function but did not investigate the treatment mechanism [15]. Choi et al. and their colleagues demonstrated that high-dose (3D spheroid cells; 1 × 10^6^) PD-MSCs were more effective to treat ovarian dysfunction in the hemi-OVX rat model than the low-dose (single cells; 1 × 10^5^) PD-MSCs. However, their results demonstrate the therapeutic effect of cell transplantation capacity in a state where the shape of the cell (e.g., single and spheroid type) is not constant, so it is insufficient to select the cell capacity for treatment [32]. Park et al. and their colleagues demonstrated that low-dose PD-MSCs were more effective in improving ovarian function in a cell-dose experiment for the comparison of therapeutic effects in an ovarian dysfunction model using thioacetamide (TAA) [33]. Their results may help with cell capacity for ovarian function improvement, butfurther studies are needed to select cell capacity in ovarian dysfunction models on various factors. Considering preclinical results comparing the number of transplanted stem cells to treat multiple degenerative diseases, our data can help determine the optimal stem cell capacity and therapeutic effects of PD-MSCs through antioxidant mechanisms in ovarian dysfunction models. Subsequently, for the development of treatments using PD-MSCs, it is essential to demonstrate therapeutic effects in single and repeated transplantation of PD-MSCs in future studies.

## 5. Conclusions

This study showed that, compared with a high concentration of transplanted stem cells, a low concentration of transplanted stem cells could restore ovarian function through the activation of antioxidant factors. Thus, our report suggests a possible appropriate dose for the transplantation of stem cells to restore ovarian function through antioxidant effects; however, further studies are needed in a wider variety of models of ovarian dysfunction. Although substantial work in clinical trials is still needed for the development of stem cells, these findings are meaningful for establishing new guidelines for MSC therapy to improve therapeutic efficacy.

## Figures and Tables

**Figure 1 antioxidants-12-01575-f001:**
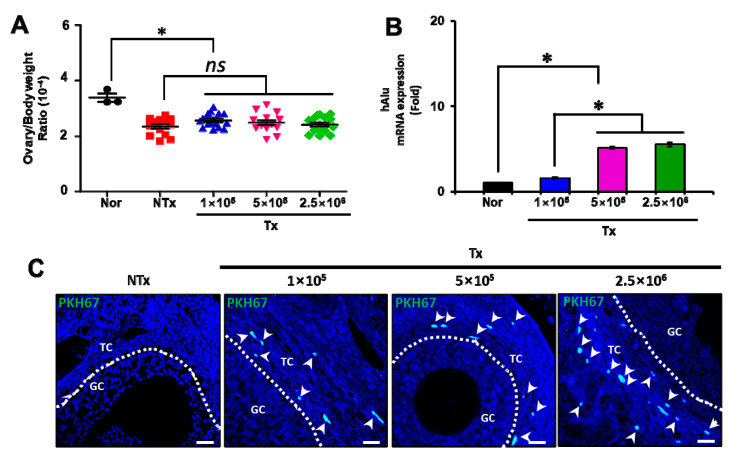
Effect of PD-MSCs on engraftment potential. The ratio of ovarian weight to body weight after sacrifice (mean ± SE) (**A**). The mRNA expression of the hAlu gene in the ovarian tissues of 1/2 OVX rats was analyzed 1 week after transplantation using qRT–PCR analysis and normalized to rat GAPDH expression, which served as the internal control (mean ± SE) (**B**). The localization of PKH-67-labeled PD-MSCs in the ovarian tissues of 1/2 OVX rats was analyzed 1 week after transplantation via IF staining. The white arrow means the PKH-67 labeled PD-MSCs (**C**). Significance at *p* < 0.05 is indicated with an asterisk (*).

**Figure 2 antioxidants-12-01575-f002:**
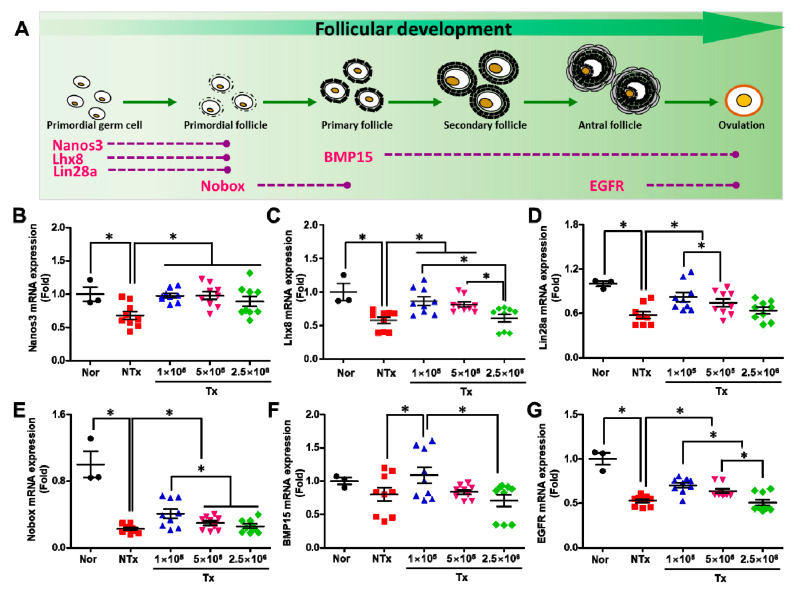
Effect of PD-MSCs on follicular development. Schematic of follicular development according to each maturation stage (**A**). The mRNA expression of Nanos3 (**B**), Lhx8 (**C**), Lin28a (**D**), Nobox (**E**), BMP15 (**F**), and EGFR (**G**) in the ovarian tissues of 1/2 OVX rats 1, 3, and 5 weeks after transplantation was analyzed using qRT–PCR (mean ± SE). Significance at *p* < 0.05 is indicated with an asterisk (*).

**Figure 3 antioxidants-12-01575-f003:**
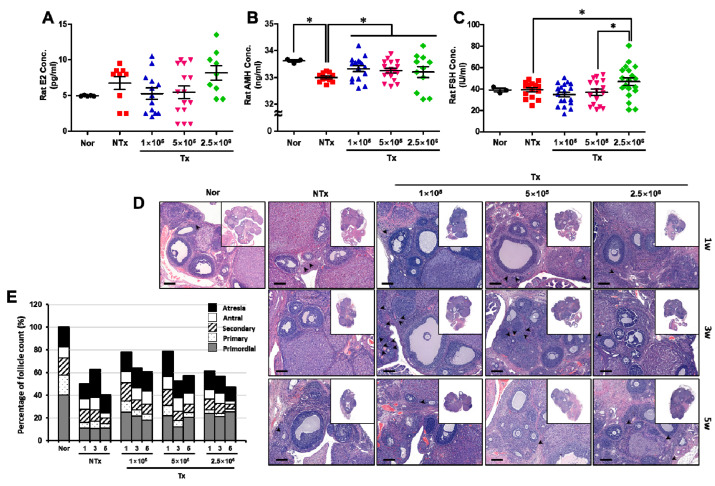
Effect of PD-MSCs on ovarian function. The levels of E2 (**A**), AMH (**B**), and FSH (**C**) in the serum of individual 1/2 OVX rats 1, 3, and 5 weeks after transplantation were analyzed via ELISA (mean ± SE). Histological analysis (**D**) and the number of follicles in the ovarian tissues of 1/2 OVX rats 1, 3, and 5 weeks after transplantation were analyzed and counted using H&E staining (mean ± SE). The black arrows indicate the primorgial follicles in ovarian tissue (**E**). Significance at *p* < 0.05 is indicated with an asterisk (*).

**Figure 4 antioxidants-12-01575-f004:**
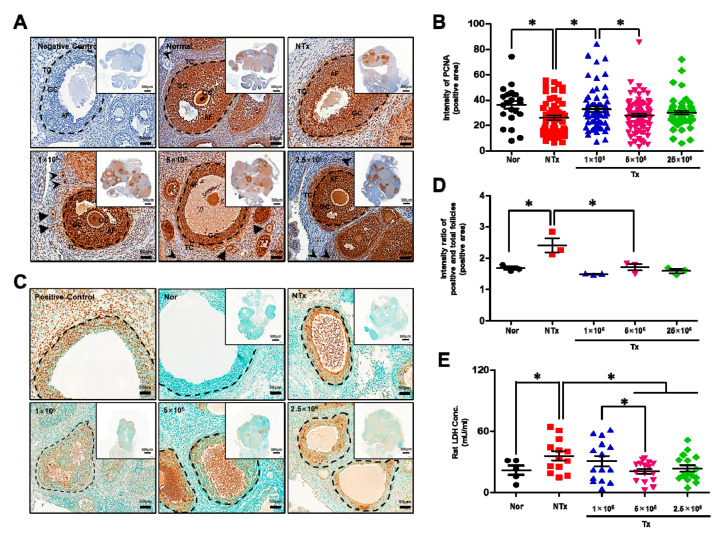
Effect of PD-MSCs on proliferation and cell death. The positive PCNA signals in the ovarian tissues of 1/2 OVX rats 1, 3 and 5 weeks after transplantation were analyzed via IHC staining. The black dotted lines indicate the follicle in ovarian tissue. The black arrow and triangles indicate the primordial and primary follicles (**A**). The staining intensity in the antral follicles of the ovarian tissues of 1/2 OVX rats 1, 3, and 5 weeks after transplantation was quantified using the Dongle program (**B**). The positive cell death signals in the ovarian tissues of 1/2 OVX rats 1, 3, and 5 weeks after transplantation were analyzed using TUNEL assay. The black dotted lines indicate the follicle in ovarian tissue (**C**). The staining intensity in the antral follicles of the ovarian tissues of 1/2 OVX rats 1, 3, and 5 weeks after transplantation was quantified using the Dongle program (**D**). The levels of LDH in the serum of 1/2 OVX rats 1, 3, and 5 weeks after transplantation were analyzed via ELISA (**E**). Significance at *p* < 0.05 is indicated with an asterisk (*).

**Figure 5 antioxidants-12-01575-f005:**
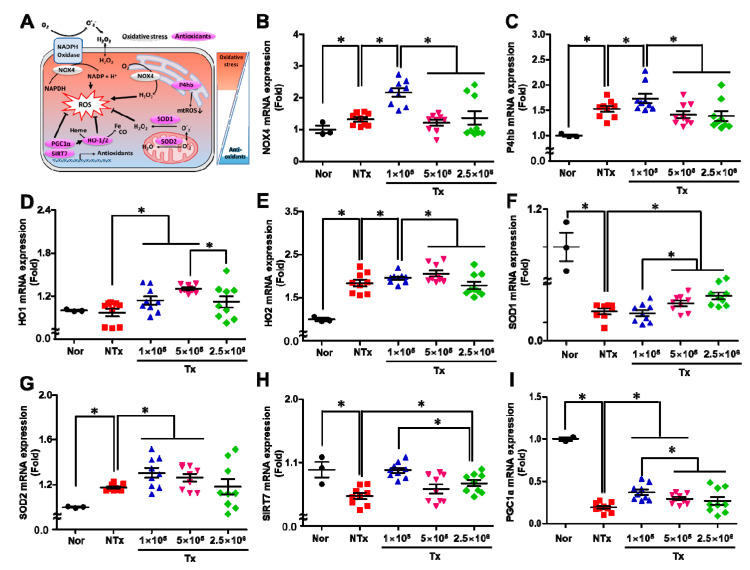
The effect of PD-MSCs on oxidative stress. Schematic of antioxidant signaling (**A**). The mRNA expression of NOX4 (**B**), P4hb (**C**), HO-1 (**D**), HO-2 (**E**), SOD1 (**F**), SOD2 (**G**), SIRT7 (**H**), and PGC1α (**I**) in the ovarian tissues of 1/2 OVX rats 1, 3 and 5 weeks after transplantation was analyzed using qRT–PCR (mean ± SE). Significance at *p* < 0.05 is indicated with an asterisk (*).

**Figure 6 antioxidants-12-01575-f006:**
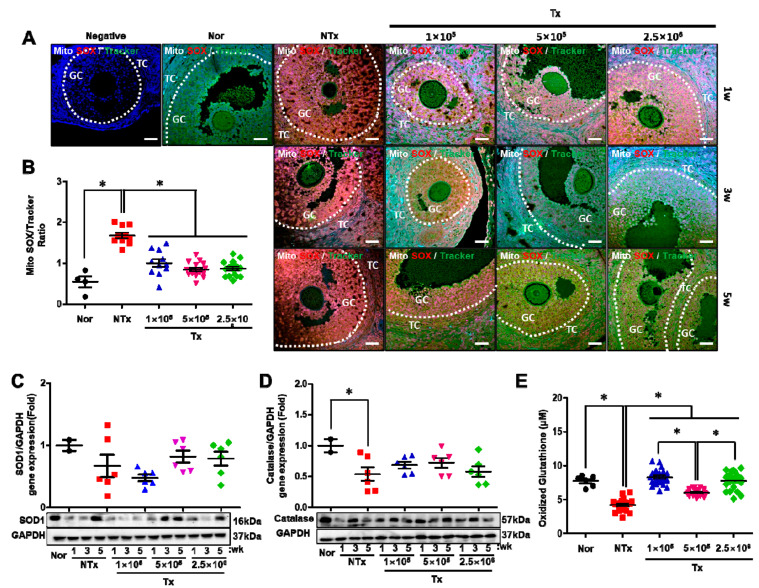
Effect of PD-MSCs on the antioxidant signaling pathway. The positive superoxide signals (i.e., MitoSOX) and mitochondria signals (i.e., MitoTracker) in the ovarian tissues of 1/2 OVX rats 1, 3 and 5 weeks after transplantation were analyzed via IF staining (**A**). The ratio of MitoSOX intensity to MitoTracker intensity in the ovarian tissues of 1/2 OVX rats 1, 3, and 5 weeks after transplantation was analyzed using the ImageJ program (mean ± SE). The white dotted lines indicate the follicles in ovarian tissue (**B**). The gene expression of SOD1 (**C**) and catalase (**D**) in the ovarian tissues of 1/2 OVX rats 1, 3 and 5 weeks after transplantation were analyzed via Western blotting (mean ± SE). The levels of oxidized glutathione in the serum of individual 1/2 OVX rats 1, 3 and 5 weeks after transplantation were analyzed using ELISA (mean ± SE) (**E**). Significance at *p* < 0.05 is indicated with an asterisk (*).

**Figure 7 antioxidants-12-01575-f007:**
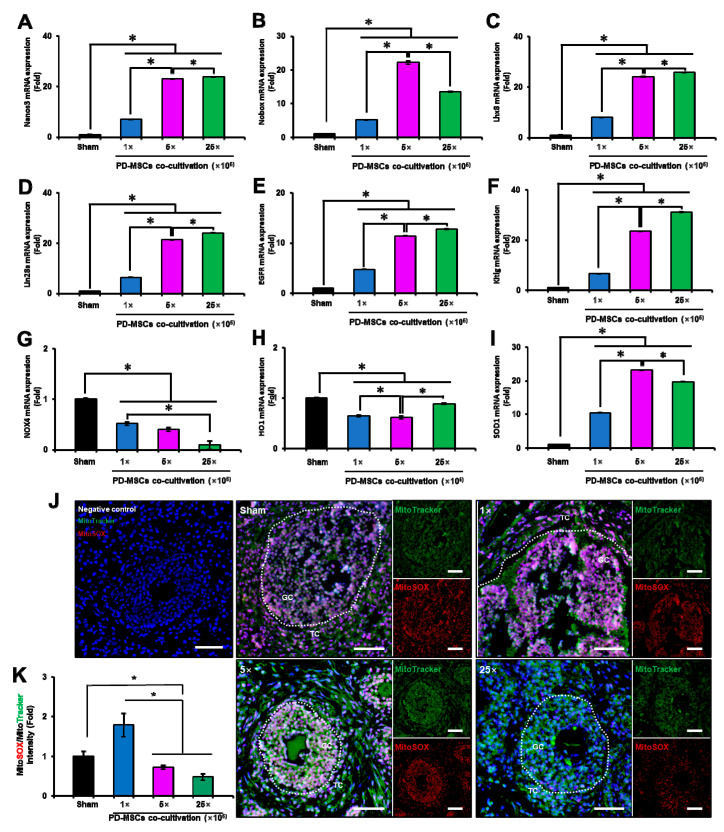
Effect of PD-MSC cocultivation for 48 h on follicular development and antioxidant activity. The mRNA expression of Nanos3 (**A**), Nobox (**B**), Lhx8 (**C**), Lin28a (**D**), EGFR (**E**),Kitlg (**F**), NOX4 (**G**), HO1 (**H**), and SOD1 (**I**) in ovarian tissues cocultured with PD-MSCs for 48 h. The positive superoxide signals (i.e., MitoSOX) and mitochondria signals (i.e., MitoTracker) in the ovarian tissues cocultured with PD-MSCs for 48 h were analyzed via IF staining. The white dotted lines indicate the follicle in the ovarian tissue (**J**). The ratio of MitoSOX intensity to MitoTracker intensity in the ovarian tissues cocultured with PD-MSCs for 48 h was analyzed (**K**). Significance at *p* < 0.05 is indicated with an asterisk (*).

**Table 1 antioxidants-12-01575-t001:** Comparison of follicle counts after transplantation of PD-MSC in vivo.

Group			Primordial (%)	Primary (%)	Secondary (%)	Antral (%)	Atresia (%)
Nor			39.92 ± 2.42	17.72 ± 0.60	14.88 ± 0.70	10.02 ± 0.77	17.46 ± 0.99
1w	NTx		22.64 ± 3.54 *	9.59 ± 0.90 *	22.93 ± 0.62 *	19.17 ± 3.73 *	25.67 ± 5.31 *
	Tx	1 × 10^5^	31.56 ± 1.48 **	13.02 ± 0.56 **	19.84 ± 1.70 **	12.21 ± 1.68	23.38 ± 4.30
		5 × 10^5^	27.25 ± 5.93 **, #	11.34 ± 0.80#	18.08 ± 2.10	14.26 ± 1.53 **	29.07 ± 6.77
		2.5 × 10^6^	46.58 ± 7.10 **, ##	6.57 ± 1.17 **, #, ##	15.49 ± 1.57 **,#	11.43 ± 2.47 **	19.93 ± 7.46
3w	NTx		17.12 ± 3.32 *	10.22 ± 3.32 *	15.81 ± 0.95	17.28 ± 0.57 *	39.57 ± 0.73 *
	Tx	1 × 10^5^	33.67 ± 0.22 **	8.62 ± 1.21	13.81 ± 1.75	16.53 ± 2.25	27.37 ± 1.55 **
		5 × 10^5^	26.39 ± 5.60 **, #	10.57 ± 1.30	14.00 ± 1.10	20.98 ± 5.32	28.05 ± 0.56 **
		2.5 × 10^6^	37.04 ± 3.25 **, ##	5.25 ± 1.47 ##	15.11 ± 3.28	15.81 ± 4.77	26.78 ± 4.86 **
5w	NTx		27.64 ± 1.19 *	9.65 ± 1.70 *	11.95 ± 1.93 *	11.32 ± 1.83	39.43 ± 6.09 *
	Tx	1 × 10^5^	29.86 ± 1.16	8.64 ± 0.73	14.23 ± 1.15	19.38 ± 3.65 **	27.89 ± 3.30 **
		5 × 10^5^	28.30 ± 7.97	9.47 ± 2.13	11.32 ± 1.99	16.07 ± 3.22	34.84 ± 8.42
		2.5 × 10^6^	53.74 ± 0.54 **, #, ##	5.62 ± 0.39 ##	8.45 ± 0.14 **, #	5.60 ± 1.13 **, #	26.59 ± 0.29 **

Nor, a normal group. NTx, non-transplantation group. Tx, PD-MSC transplantation group. * Significantly different versus the normal group (* *p* < 0.05). ** Significantly different versus NTx (** *p* < 0.05). # Significantly different versus 1 × 10^5^ Tx (# *p* < 0.05). ## Significantly different versus 5 × 10^5^ Tx (## *p* < 0.05).

## Data Availability

All data generated or analyzed during this study are included in this manuscript and its Appendix A.

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
