# Peer review of "The Dose-Related Efficacy of Human Placenta-Derived Mesenchymal Stem Cell Transplantation on Antioxidant Effects in a Rat Model with Ovariectomy"

_antioxidants, 2023, doi:10.3390/antiox12081575_

Round 1
Reviewer 1 Report
In the manuscript entitled “Comparative analysis of the dose and efficacy of stem cell transplantation on antioxidant effects in a rat model with ovarian insufficiency” by Jin Seok et al., the Authors reported the effects of placenta-derived mesenchymal stem cell transplantation in ovarian function in half-ovariectomized rats by exerting antioxidant effects. The topic is interesting but there are some issues that must be revised by the Authors. In particular:
Major points
1. Fluorescence images in Fig. 1C should display PKH-67 green fluorescence in the labeled cells, but the green signal in the cytoplasm is absent. The merge with DAPI should be added after showing the individual signals.
2. Fluorescence images Fig. in 7J have poor quality. The merge images should be added after showing the individual signals. Moreover, a higher magnification (in addition to the magnification shown) would be useful to better appreciate the signal specificity.
3. Why do Nor groups show only 3 samples compared with the other groups?
Author Response
Author's Reply to the Review Report (Reviewer 1)
In the manuscript entitled “Comparative analysis of the dose and efficacy of stem cell transplantation on antioxidant effects in a rat model with ovarian insufficiency” by Jin Seok et al., the Authors reported the effects of placenta-derived mesenchymal stem cell transplantation in ovarian function in half-ovariectomized rats by exerting antioxidant effects. The topic is interesting but there are some issues that must be revised by the Authors. In particular:
Author’s response:
Thank you for your positive comments with considerable review. Responses to the comments were listed as follows.
Major points
- 1. Fluorescence images in Fig. 1C should display PKH-67 green fluorescence in the labeled cells, but the green signal in the cytoplasm is absent. The merge image with DAPI should be added after showing the individual signals.
Author’s response #1:
Thank you for your comment. We replaced it by adding individual signals of PKH-67 green fluorescence in Figure 1C of the revised manuscript.
- 2. Fluorescence images Fig. in 7J have poor quality. The merge images should be added after showing the individual signals. Moreover, a higher magnification (in addition to the magnification shown) would be useful to better appreciate the signal specificity.
Author’s response #2:
Thank you for your comment. We replaced it by adding the individual signals of MitoSOX and MitoTracker in Figure 7J of the revised manuscript. Also, we replaced it with images with high magnification as Reviewer requested.
- 3. Why do Nor groups show only 3 samples compared with the other groups?
Author’s response #3:
Thank you for your critical comment. In the process of analyzing various comparison groups (e.g., various ovarian dysfunction animal models, transplant numbers, and transplant routes), we secured a sufficient number of individuals for the test group (n<10), but the normal group consisted of the minimum number (n=3) that can be analyzed statistically to identify the efficacy of stem cells.

Reviewer 2 Report
The present paper by J. Seok et al. deals with the injection of human placental stem cells to overcome ovarian insufficiency in a chirurgically-obtained rat model.
General comment :
My main concern with this paper is that I do not understand the model : rats with two half-removed ovaries (not hemi-ovariectomized as I understood in the first place). The rationale for using this very particular model should be explained more clearly.
Why not also test rats with one half-removed ovary at one side and keep the other one just cut in two but left in place for comparison in the same animal ? In brief, the model should be more clearly defined and justified in the scope of the study (not just by references 9, 12, 13).
In some instances, it was indicated that ovarian function was studied in ovariectomized (OVX) rats ; that sounds pretty weird if it means that the two ovaries are removed. The models used should be more clearly explained.
Moreover, there must be some compensatory growth of the remaining half-ovaries. How can it be taken into account ?
Specific comments :
1/ The title might be shortened and more precise ; maybe : « Dose-related efficacy of human placental stem cells… ». Also the mention of « a rat model with ovarian insufficiency » is too vague.
2/ Lines 28-30 : Isn’t it contradictory to say that after cell transplantation, there are anti-oxidant effects whereas oxidized glutathione was increased ?
3/ Line 41 : ROS are strongly …
4/ Line 47 : it would better to mention glutathione oxidoreductase (and/or other enzymes) which is an enzyme like SOD and catalase rather than only glutathione.
5/ Line 95 : UC-MSC should be defined at this first occurrence of UC.
6/ Line 106-107 : add reference
7/ Line 117 : It should be mentioned that the cells from the chorionic plate of placental tissues are cells from the fetal side of the placenta. How homogeneous (or not) are these cells ?
8/ Line 253 : « ½ of OVX rats » not clear.
9/ Lines 253-4 : How can it be to experience more severe ovarian failure than OVX ? What exactly means OVX in this paper ?
10/ Figure 3E : It seems that the injection of PD-MSC essentially increased the number of primordia follicles, not the other categories.
11/ Table : for clarity indicate « …. transplantation of PD-MSC…. » in the table’s title.
12/ Lines 470-485 : it would have been interesting to test the supernatant of cultured PD-MSC in parallel to the coculture to determine whether secreted soluble factors are involved.
13/ Line 578 : « treat ovarian dysfunction in the OVX model » sounds strange. What is called « OVX rat model ? » precisely in this paper ?
Author Response
Author's Reply to the Review Report (Reviewer 2)
The present paper by J. Seok et al. deals with the injection of human placental stem cells to overcome ovarian insufficiency in a chirurgically-obtained rat model.
General comment :
My main concern with this paper is that I do not understand the model : rats with two half-removed ovaries (not hemi-ovariectomized as I understood in the first place). The rationale for using this very particular model should be explained more clearly.
Why not also test rats with one half-removed ovary at one side and keep the other one just cut in two but left in place for comparison in the same animal ? In brief, the model should be more clearly defined and justified in the scope of the study (not just by references 9, 12, 13).
In some instances, it was indicated that ovarian function was studied in ovariectomized (OVX) rats ; that sounds pretty weird if it means that the two ovaries are removed. The models used should be more clearly explained.
Moreover, there must be some compensatory growth of the remaining half-ovaries. How can it be taken into account ?
Author’s response:
Thank you for your critical comments with considerable review.
Ovariectomized rat model (OVX) has been widely used to study degenerative diseases caused by female reproductive hormones by removing both ovaries in general. However, the function of the ovary could not be studied because the ovary was removed in OVX model.
Accordingly, we studied hormonal changes and the function of the preserved ovary after removing only one of the two ovaries and preserving the other ovary.
In this study, we generated a model that induced physical damage by removing only half of both ovaries. We confirmed that ovarian function significantly decreased at the level of mRNA expression in the 1/2 ovariectomized model (1/2 OVX), in which only half of both ovaries were removed, compared to the OVX model, in which only one ovary was removed. So, we suggest that the 1/2 ovariectomized (1/2 OVX) model could be a severe ovarian dysfunction model as can analyze ovarian function.
To prevent confusion among readers, we revised the model with only one ovary removed to hemi-OVX and the model with only half of both ovaries removed to 1/2 OVX in the revised manuscript.
Specific comments :
1/ The title might be shortened and more precise; maybe : « Dose-related efficacy of human placental stem cells… ». Also the mention of « a rat model with ovarian insufficiency » is too vague.
Author’s response #1:
As you commented, the title was corrected to " Dose-related efficacy of human placenta-derived mesenchymal stem cell transplantation on antioxidant effects in a rat model with ovariectomy" in the revised manuscript.
2/ Lines 28-30 : Isn’t it contradictory to say that after cell transplantation, there are anti-oxidant effects whereas oxidized glutathione was increased ?
Author’s response #2:
Thank you for your critical comment. Generally, glutathione has a main role not only in allowing disulfide bonds to form but also in balancing redox reactions and thereby protecting the cell from oxidative stress. So, glutathione is oxidized during peroxide disposal by glutathione peroxidase to GSSG (two molecules of disulfide-bonded GSH) and is regenerated from GSSG by glutathione reductase in the cells. In addition, GSF can react in glutathione-S-transferase-catalyzed reactions with xenobiotics and endogenous compounds. Therefore, transplanting stem cells could induce to increase in oxidative stress in diseases model via the oxidization of glutathione for balancing redox reactions in the body.
References:
- The role of glutathione in disulfide bond formation and endoplasmic-reticulum-generated oxidative stress. EMBO Rep. 2006 Mar; 7(3): 271–275.
- Chapter Five - The antioxidant glutathione. Vitamins and Hormones Volume 121, 2023, Pages 109-141
3/ Line 41 : ROS are strongly …
Author’s response #3:
Sorry for the confusion caused by spelling errors. We corrected it in the revised manuscript.
4/ Line 47 : it would better to mention glutathione oxidoreductase (and/or other enzymes) which is an enzyme like SOD and catalase rather than only glutathione.
Author’s response #4:
Thank you for your kind correction. We revised it in the revised manuscript.
5/ Line 95 : UC-MSC should be defined at this first occurrence of UC.
Author’s response #5:
Thank you for your kind suggestion. We added the full name of UC-MSCs in the revised manuscript.
6/ Line 106-107 : add reference
Author’s response #6:
Thank you for your kind suggestion. We added references in the revised manuscript.
7/ Line 117 : It should be mentioned that the cells from the chorionic plate of placental tissues are cells from the fetal side of the placenta. How homogeneous (or not) are these cells ?
Author’s response #7:
Thank you for your kind suggestion. We added the origin of the chorionic plate membrane isolated from the fetal side of the normal-term placenta in the revised manuscript. Especially, the lineage of these cells is mesodermal origin among three-germ lineages (endodermal, ectodermal, and mesodermal lineages). Also, only mesenchymal stem cells exist without any other cell types in the chorionic plate membrane.
8/ Line 253 : « ½ of OVX rats » not clear.
9/ Lines 253-4 : How can it be to experience more severe ovarian failure than OVX ? What exactly means OVX in this paper ?
Author’s response #8,9:
Thank you for your critical point. Please see the response to question 1 for Reviewer 1.
In brief, ovariectomized rat model (OVX) has been widely used to study degenerative diseases caused by female reproductive hormones by removing both ovaries in general. However, the function of the ovary could not be studied because the ovary was removed in OVX model. Accordingly, we studied hormonal changes and the function of the preserved ovary after removing only one of the two ovaries and preserving the other ovary.
In this study, we generated a model that induced physical damage by removing only half of both ovaries. We confirmed that ovarian function significantly decreased at the level of mRNA expression in 1/2 ovariectomized model (1/2 OVX), in which only half of both ovaries were removed, compared to OVX model, in which only one ovary was removed. So, we suggest that 1/2 ovariectomized (1/2 OVX) model could be a severe ovarian dysfunction model as can analyze ovarian function.
To prevent confusion among readers, we revised the model with only one ovary removed to hemi-OVX and the model with only half of both ovaries removed to 1/2 OVX in the revised manuscript.
10/ Figure 3E: It seems that the injection of PD-MSC essentially increased the number of primordia follicles, not the other categories.
Author’s response #10:
In your opinion, only primordial follicles may appear to be increased. The results of Figure 3E confirmed that the overall level excluding atresia follicles increased in the group transplanted with stem cells. Stem cell transplantation into ovarian-damaged models is thought to maintain the number of primordial follicles by inhibiting injury-induced cell death in ovarian tissues. Also, as shown in Table 1, an increase in the number of follicles including primary and secondary is also observed. It was confirmed that the number of antral follicles significantly increased 5 weeks after stem cell transplantation.
11/ Table : for clarity indicate « …. transplantation of PD-MSC…. » in the table’s title.
Author’s response #11:
Thank you for your kind correction. We revised the table title in the revised manuscript.
12/ Lines 470-485 : it would have been interesting to test the supernatant of cultured PD-MSC in parallel to the coculture to determine whether secreted soluble factors are involved.
Author’s response #12:
Thank you for your important comment. Although we did not include this information in this manuscript, the feasibility of cytokines (e.g., stem cell factor; SCF) secreted by PD-MSCs could be needed in the near future.
13/ Line 578 : « treat ovarian dysfunction in the OVX model » sounds strange. What is called « OVX rat model ? » precisely in this paper ?
Author’s response #13:
Thank you for your comment. In the study, we used OVX model, which removed only one ovary and retained the other ovary. To prevent confusion among readers, we revised the model with only one ovary removed to hemi-OVX and the model with only half of both ovaries removed to 1/2 OVX in the revised manuscript.

Round 2
Reviewer 1 Report
No further comments
Reviewer 2 Report
The authors have adequately answered the issues I raised.
Thank you